# Betanin Attenuates Epigenetic Mechanisms and UV-Induced DNA Fragmentation in HaCaT Cells: Implications for Skin Cancer Chemoprevention

**DOI:** 10.3390/nu16060860

**Published:** 2024-03-16

**Authors:** Afshin Zand, Sodbuyan Enkhbilguun, John M. Macharia, Krisztina Varajti, Istvan Szabó, Gellért Gerencsér, Boglárka Bernadett Tisza, Bence L. Raposa, Zoltán Gyöngyi, Timea Varjas

**Affiliations:** 1Department of Public Health Medicine, Medical School, University of Pecs, 7624 Pecs, Hungary; bokor28@yahoo.co.uk (I.S.); gellert.gerencser@gmail.com (G.G.); zoltan.gyongyi@aok.pte.hu (Z.G.); timea.varjas@aok.pte.hu (T.V.); 2Faculty of Health Sciences, University of Pecs, 7621 Pecs, Hungary; enkhbilguunsod@gmail.com (S.E.); raposa.laszlo@pte.hu (B.L.R.); 3Doctoral School of Health Sciences, Faculty of Health Science, University of Pecs, 7621 Pecs, Hungary; johnmacharia@rocketmail.com (J.M.M.); boglarka.tisza@etk.pte.hu (B.B.T.); 4Doctoral School of Clinical Medicine, Medical School, University of Pecs, 7624 Pecs, Hungary; varajti.krisztina@gmail.com

**Keywords:** DNMT, HDAC, DNA fragmentation, betanin, beetroot, skin cancer, qRT–PCR, gene expression, ultraviolet exposure, comet assay, XTT assay

## Abstract

Dermal photoaging refers to the skin’s response to prolonged and excessive ultraviolet (UV) exposure, resulting in inflammation, changes to the tissue, redness, swelling, and discomfort. Betanin is the primary betacyanin in red beetroot (*Beta vulgaris*) and has excellent antioxidant properties. Yet, the specific molecular mechanisms of betanin in HaCaT cells have not been fully clarified. The objective of this study was to investigate the activity of betanin and the underlying mechanisms in HaCaT cells; furthermore, in this study, we explored the protective effect of various concentrations of betanin against UVB irradiation on HaCaT cells. Additionally, we assessed its influence on the transcription of various epigenetic effectors, including members of the DNA methyltransferase (*DNMT*) and histone deacetylase (*HDAC*) families. Our findings demonstrate a notable downregulation of genes in HaCaT cells, exhibiting diverse patterns upon betanin intake. We considered the involvement of *DNMT* and *HDAC* genes in distinct stages of carcinogenesis and the limited exploration of the effects of daily exposure dosages. Our results indicate that betanin may protect the skin from damage caused by UV exposure. Further investigation is essential to explore these potential associations.

## 1. Introduction

Solar ultraviolet radiation (UVR) has a significant environmental influence on human skin. It causes various harmful effects, such as sunburn, redness, swelling, weakened immunity, and the development of skin cancers [1]. Studies tracking populations have confirmed a link between prolonged or intense exposure to sunlight and a greater likelihood of developing melanoma and nonmelanoma skin cancers [2]. The ozone layer filters out most solar radiation below 290 nm, leaving about 5–10% UVB (290–320 nm) and 90–95% UVA (320–400 nm) reaching the skin. Both UVA and UVB can damage DNA in mammalian cells, with UVB causing CPDs and (6–4) PPs [3,4]. 

Moreover, when exposed to UVB radiation, human skin triggers the release of reactive oxygen species (ROS), activating various signaling pathways in keratinocytes and fibroblasts [5,6]. Unrepaired DNA lesions from UVB radiation can lead to cell death or mutations, promoting skin tumor development. UVB damage triggers reactive oxygen species (ROS) production and reduces endogenous antioxidants in the skin [7,8,9]. In mature skin, reduced antioxidant enzyme expression in the stratum corneum and epidermis leads to oxidatively modified proteins accumulating in the upper dermis of photoaged skin [10]. Hence, the pivotal role of UVB-induced oxidative stress underscores the importance of employing intrinsic and extrinsic antioxidants to alleviate its impact.

Beetroot (*Beta vulgaris* L.), a member of the Chenopodiaceae family, is recognized as a valuable reservoir of essential bioactive compounds, such as betalains, nitrate (NO_3_^−^), and phenolic compounds [11,12].

The primary component of red beetroot pigments is betanin (Figure 1), which exhibits various beneficial biological effects, including antioxidative, anti-inflammatory, hepatoprotective, and antitumor activities [13,14]. Studies have shown that beetroot and its compound betanin act as strong chemopreventive agents. They induce apoptosis, decrease cell proliferation, inhibit angiogenesis, and reduce inflammation in animal models and cancer cell lines linked to skin, liver, lung, and esophageal cancers [15,16,17,18].

Furthermore, additional chemopreventive effects of betanin have been documented, including a sustained delay in tumor onset, an extension of tumor latency, and a reduction in tumor multiplicity [19,20]. Various studies have demonstrated that betanin, the principal component of red beetroot, exhibits anticancer and antiproliferative activities by stimulating antioxidant defense and scavenging lipoperoxyl free radicals [21,22,23].

Moreover, betanin may play a significant role in preventing injuries and cancers in the liver, lungs, and kidneys through the stimulation of detoxifying and antioxidant enzyme expression and the reduction of oxidative stress induced by xenobiotics [24,25,26,27]. Based on earlier research, beetroot and its primary component, betanin, and other compounds possessing potent antioxidant, antiproliferative, and antitumor activities may be efficacious during cancer initiation and promotion [19,24].

The comet assay, recognized for its reliability, is a reliable method for assessing DNA damage. Its advantages include simplicity, sensitivity, capacity to yield quantitative results, rapidity, and cost-effectiveness. This method can be applied to any eukaryotic cell to ascertain genotoxicity [28,29,30]. 

DNA methyltransferases (DNMTs) and histone deacetylases (HDACs) are critical players in epigenetics. DNMTs maintain genomic stability by methylating cytosine residues at the 5-position in DNA, regulating gene expression by silencing genes through the methylation of promoter regions, hindering transcription factor binding [31]. DNMTs, including DNMT1, DNMT3A, and DNMT3B, play crucial roles in DNA methylation. The dysregulation of these enzymes can lead to chromosomal instability and promote carcinogenesis by aberrant DNA methylation. The overexpression of DNMTs may silence tumor-suppressor genes, potentially fostering cancer development [32]. The excessive methylation of DNA repair gene promoters is strongly linked to various human tumor types, including colon, breast, and lung cancers [33].

HDACs are enzymes that remove acetyl groups from histones, impacting gene expression. Research has shown their involvement in altering acetylation levels, particularly in various cancer cells. The significant role of HDACs in initiating, advancing, and promoting carcinogenesis is well-established [34].

HDAC5 and HDAC6, among other HDAC enzymes, play vital roles in regulating cell proliferation. They interact with transcriptional repressors and co-repressors, such as polycomb group proteins, to suppress genes involved in cell proliferation. Studies suggest that HDACs hinder cell cycle inhibitors, differentiation, and apoptosis while facilitating angiogenesis, invasion, and migration [35,36].

This study aimed to assess the association between betanin and the expression levels of *HDAC2*, *HDAC3*, and *HDAC8*, along with *DNMT1*, *DNMT3a*, and *DNMT3b*. This evaluation was conducted using qRT-PCR on HaCaT cells exposed to varying durations of UVB radiation.

## 2. Materials and Methods

### 2.1. Cell Culture

HaCat human immortalized keratinocyte cell lines (Cytion catalog number 300493, Eppelheim, Germany) were cultured in Dulbecco’s modified Eagle’s medium (Merck-Sigma-Aldrichc (Budapest, Hungary), catalog number D6429) supplemented with 10% Fetal Bovine Serum (Biosera, Cholet, France) and 1x penicillin/streptomycin solution (Thermo Fisher scientific, catalog number 15140122). Cells were incubated at 37 °C in an environment with 5% CO_2_ [37]. Following the manufacturer’s protocol, HaCaT cells were seeded at 1.0 × 10^4^ cells per cm^2^. Seeding cell concentration was measured by the TC20 automated cell counter (Bio-Rad Inc., Hercules, CA, USA). Treatment was carried out when reaching 40–50% confluence.

### 2.2. UV Radiation and Betanin Treatment

Following a 24 h incubation period, the culture medium was replaced with phosphate-buffered saline (PBS) (Thermo Fisher Scientific Inc., Waltham, MA, USA). Subsequently, the cells were subjected to UVB radiation (125 mJ/cm^2^) radiation, employing a Bio-Link BLX-312 UVB lamp (Vilber Lourmat GmbH, Marne-la-Vallée, France) for varying durations (15, 30, and 60 s). Following UV exposure, the PBS was replaced with a fresh culture medium containing different concentrations of betanin (Merck-Sigma-Aldrich, Budapest, Hungary) (0 µM, 20 µM, 40 µM, and 80 µM), (Table 1), and the cells were treated for an additional 24 h. After this incubation period, the betanin-containing medium was removed, and the cells were washed with PBS. Cell harvesting was achieved by trypsinization using a solution containing 0.05% Trypsin/EDTA (Thermo Fisher Scientific Inc., Waltham, MA, USA) and 0.01% EDTA (Sigma–Aldrich, Budapest, Hungary) Subsequently, cell supernatants were collected for further analysis.

### 2.3. Comet Assay Evaluation

The experiment involved the application of three layers of gel on microscope slides. The first layer, a standard melting point agarose (NMA) (Merck-Sigma-Aldrich, Budapest, Hungary) with a concentration of 0.5%, was followed by a middle layer and a covering layer of low-melting-point agarose (LMPA) (Merck-Sigma-Aldrich, Budapest, Hungary) at a 1% concentration. The keratinocyte cells were suspended in the middle LMPA layer. The slides were kept in a lysing solution (Merck-Sigma-Aldrich, Budapest, Hungary) in the dark at 4 °C for 24 h. The nuclear DNA was electrophoresed under alkaline conditions to partially disrupt the secondary structure and enable the movement of DNA fragments in the agarose gel. The slides were immersed in an electrophoresis buffer (200 mM EDTA (Merck-Sigma-Aldrich, Budapest, Hungary), 10 mM NaOH (Molar Chemicals Kft, Budapest, Hungary) at pH 10, for 20 min before electrophoresis was performed for 40 min at 0.46 V/cm and 132 mA in total darkness. Afterward, the slides were neutralized three times with a neutralizing buffer containing 0.4 M Tris (Merck-Sigma-Aldrich, Budapest, Hungary) for five minutes each and then stained with ethidium bromide (50 μL–2 μg/mL) (Merck-Sigma-Aldrich, Budapest, Hungary) [38]. The images representing DNA damage were visualized using fluorescence microscopy [39]. Comet Assay IV Imaging Software, Version 4.3.1 (Perceptive Instruments Ltd., Bury St Edmunds, UK, 400×) was used to analyze fifty nuclei on each slide (in Comet Assay IV, the level of DNA damage was assessed using the tail moment (TM), computed as the product of the tail length and the percentage of DNA present in the tail).

### 2.4. Cell Viability/XTT Assay

The photoprotective impact of betanin against UVB radiation on HaCaT cells was assessed using the XTT assay. The CyQUANT XTT Cell Viability assay kit (Thermo Fisher) was employed. HaCaT cells were seeded onto 96-well plates at appropriate densities and incubated for the specified duration. Following this, they underwent two washes with PBS. The control group remained untreated, while the model group was subjected to UVB irradiation for different durations (0, 15, 30, and 60 s), and the drug administration group received varying concentrations of betanin solution (0, 20, 40, and 80 μM). Post-administration, HaCaT cells were incubated for 24 h. Subsequently, 70 μL of an XTT working solution (a mixture of XTT Reagent and Reagent) was added to each well, and the cells were further incubated at 37 °C. After 4 h of incubation, absorbance was measured at 450 and 660 nm wavelengths using an absorbance Dialab microplate reader (Budapest, Hungary). The calculation of the specific absorbance of the sample was conducted utilizing the following formula:Specific Absorbance = [Abs450 nm(Test) − Abs450 nm(Blank)] − Abs660 nm(Test)

### 2.5. RNA Extraction and Quantitative Real-Time PCR Analysis

Total RNA was isolated via a semiautomatic workflow with the help of the Maxwell^®^ RSC Instrument (Promega, Fitchburg, WI, USA) using a Maxwell^®^ RSC RNA FFPE Kit (AS1440, Promega, Fitchburg, WI, USA). The process was conducted according to the manufacturer’s instructions. The quality of isolated RNA was assessed utilizing the Thermo Scientific NanoDrop^TM^ 2000 (Thermo Fisher Scientific, Grand Island, NY, USA). RNA was eluted in 50 µL of RNase-free water in the final step.

The quantitative real-time PCR assays were performed on a Roche 480 (Roche, Basel, Switzerland) instrument utilizing the KAPA SYBR FAST One-Step qRT–PCR Master Mix Kit (Sigma, Budapest, Hungary). Each reaction had a volume of 20 μL, comprising 5 μL of an RNA target (50–100 ng) and 15 μL of a master mix, which included forward and reverse primers (10 µL of KAPA SYBR FASTqPCR Master Mix, 0.4 µL of KAPA RT Mix, 0.4 µL of dUTP, 0.4 µL of primers (200 nM), and 3.8 µL of sterile double-distilled water). The amplification process involved a thermal profile with the following steps: 5 min at 42 °C for reverse transcription and a hot-start denaturing step at 95 °C for 3 min. This procedure was succeeded by 45 cycles of denaturation at 95 °C for 10 s and annealing/extension at 60 °C for 20 s. The emitted fluorogenic signal during the annealing–extension step was recorded and analyzed using software. After amplification, a melting curve was generated by increasing the temperature by 0.5 °C per cycle, starting from the setpoint temperature (55.0 °C) and repeating this process for 80 cycles, each lasting 10 s. The housekeeping gene hypoxanthine–guanin phosphoribosyltransferase 1 (*HPRT1*) was used as an internal control alongside the genes of interest *DNMT1*, *DNMT3A*, *DNMT3B*, *HDAC5*, and *HDAC6*, as shown in (Table 2). The primary sequences of these genes and the primers used for the experiment were designed using Primer Express™ 3.0.1 Software (Applied Biosystems, Budapest, Hungary) and synthesized by Integrated DNA Technologies (Bio-Sciences, Budapest, Hungary). The results were analyzed using the relative quantification (2^−∆∆CT^) method.

### 2.6. Statistical Methods

We conducted various statistical tests to evaluate our findings, including ANOVA, Levene’s F test, and subsequent post hoc analyses using the Scheffe and LSD tests. The Kolmogorov–Smirnov test was also employed to ascertain the distribution and standard deviation. The Mann–Whitney and Kruskal–Wallis tests were utilized for the comet assay data for statistical analysis. All statistical analyses were performed using IBM SPSS Statistics Version 26.0 for Windows (Armonk, NY, USA), with a significance set at *p* < 0.05 at 95% confidence interval.

## 3. Results

### 3.1. Betanin Enhances the Viability of HaCaT Subsequent to Exposure to UVB Radiation

The XTT method was utilized to investigate the toxic impacts of various Betanin concentrations on HaCaT cells. The influence of different RES concentrations (0, 10, 20, 40, and 80 μM) on HaCaT cell viability was evaluated. The findings revealed that Betanin demonstrated no cytotoxic effects on HaCaT cells at concentrations equal to or below 80 μM (Figure 2). To evaluate the potential protective influence of betanin on HaCaT cells, we conducted assays on HaCaT cells following UVB irradiation treatment using the XTT method. As depicted in Figure 2, the cell viability of HaCaT cells noticeably declined following UVB exposure. Following a 15 s UVB radiation exposure, the HaCaT cells’ viability notably decreased. However, betanin at concentrations of 20, 40, and 80 μM increased the viability of HaCaT cells by 13%, 25%, and 27%, respectively, compared to the control group.

After undergoing a 30 s UVB radiation exposure, the HaCaT cell viability significantly dropped to 60%. Nevertheless, betanin at a concentration of 80 μM notably enhanced the HaCaT cell viability by nearly 25% compared to the model group. Following a 60 s UVB radiation exposure, the HaCaT cells’ viability significantly decreased to 50%. However, betanin at concentrations of 60 μM and 80 μM notably improved the HaCaT cell viability by approximately 15% and 33%, respectively, compared to the untreated group.

### 3.2. Betanin Suppresses the Impact of UVB Radiation on DNMT Gene Expression on HaCaT

We assessed the relative expression of mRNA levels of epigenetic-associated enzymes, namely, DNMT1, DNMT3A, and DNMT3B, in immortalized human keratinocyte HaCaT cells. The relative RNA expression was quantified using *HPRT1* as a normalization reference gene. The results presented in Figure 3, Figure 4 and Figure 5 illustrate the relative gene expression patterns after exposure to varying durations of UV radiation (0, 15, 30, and 60 s), followed by the administration of betanin at diverse concentrations ranging from 20 to 80 µM.

Significant differences in expression were observed following 15, 30, and 60 s of UV radiation compared to the untreated controls (0 µM betanin). This observation underscores the evident detrimental impact of these mutations, as increased expression of DNMT genes is unfavorable for cellular proliferation.

Concerning *DNMT1*, following a 60 s radiation exposure, elevated concentrations of betanin (40 and 80 µM) demonstrated superior efficacy in conferring photoprotection compared to the control condition (Figure 3). This superiority was evidenced by the notably diminished relative expression levels, where the reduction transitioned from more than nine-fold to more than four-fold and more than one-fold, respectively. A similar discernible trend was observed after 30 s of radiation exposure, where noteworthy reductions manifested from more than six-fold to more than four-fold and more than one-fold, respectively.

Based on the findings derived from the analysis of *DNMT3A*, betanin administration yielded a discernible photoprotective influence. Nevertheless, statistical significance was exclusively attained when the exposure time was extended to 60 s of radiation combined with subsequent treatment involving 80 µM betanin compared to the untreated control (Figure 4).

In contrast, concerning *DNMT3B*, there was a substantial reduction after 15 and 30 s of radiation exposure, specifically within the purview of the 40 and 80 µM treatment regimens (Figure 5). This outcome suggested that this particular gene might elicit an early response to betanin treatment even when subjected to brief periods of radiation exposure.

### 3.3. Betanin Exhibits the Potential to Mitigate the UVB-Radiation-Induced Overexpression of HDAC Genes in HaCaT

Within the context of *HDAC5*, a marginal protective influence imparted by betanin treatment was discernible. However, the outcomes need to attain a verified level of statistical significance. Conversely, notable augmentation in expression levels after 15 and 60 s of UV radiation exposure was conspicuously evident in the untreated (0 µM betanin) control (Figure 6).

In the case of *HDAC6*, betanin consistently exhibited a robust photoprotective effect across all the radiation durations investigated. Notably, treatment with a concentration of 40 µM yielded a substantial reduction from more than 35-fold to more than 20-fold after 15 s of radiation exposure and from more than 55-fold to more than 35-fold after 60 s of radiation exposure. Moreover, treatment with a concentration of 80 µM induced a significant reduction in expression levels from more than 35-fold to more than 15-fold following 30 s of radiation and from more than 55-fold to more than 25-fold following 60 s of radiation (Figure 7). Notably, even at a concentration of 20 µM, betanin evoked a significant decrease in the relative expression level after 30 s of radiation exposure, which decreased from more than 45-fold to more than 20-fold. These cumulative findings collectively indicate that *HDAC6* is a potential candidate for inclusion in the cohort of early gene response mechanisms associated with betanin treatment.

In summary, the results showed that betanin affected the activity of all the genes tested in the *HDAC* and *DNMT* families, demonstrating its photoprotective effect. In addition, *DNMT3B* and *HDAC6* might be early-response genes to betanin treatment even after short periods of radiation.

### 3.4. The Protective Impact of Betanin against DNA Fragmentation

In our experiments, we observed that the number of DNA lesions (tail moments) in HaCaT cells significantly increased after exposure to UV radiation: two-fold and nearly four-fold after 15 s and 60 s, respectively. Treatment with betanin dose-dependently ameliorated tail-moment growth under UV irradiation for 15 s, UV irradiation for 30 s, and UV irradiation for 60 s. The 80 µM betanin treatment group detected the most significant reduction compared to the 30 s positive control group. However, the 20, 40, and 80 µM betanin treatments significantly decreased the tail moments in a dose-dependent manner after 60 s of UV irradiation, which was the most significant increase (Figure 8).

## 4. Discussion

The growing attention towards “functional foods” and their role in promoting health and managing diseases stems from the recognized benefits of diets abundant in fruits and vegetables. B. vulgaris, more commonly called red beetroot, has recently surged in popularity as a functional food renowned for its health-promoting properties [40]. Betanin, an element found in beetroot, possesses numerous advantageous physiological effects [41]. Betanin has attracted attention for its anti-inflammatory, antioxidant, and hepato-protective effects on human hepatocytes [42]. In this investigation, we observed that betanin demonstrated non-toxicity towards HaCaT cells at concentrations equal to or below 80 μM. Additionally, betanin at concentrations of 20, 40, and 80 μM exhibited a positive impact on reducing the viability of UVB-exposed HaCaT cells, thereby promoting cell proliferation and providing a protective effect. To delve into the mechanism underlying the inhibition of apoptosis by RES in vitro, we employed an RT-qPCR assay to assess the mRNA expression levels of DNMTs and HDACs. Our findings revealed that betanin pretreatment led to a reduction in the mRNA expression levels of the aforementioned genes in vitro.

### 4.1. Betanin Modulation of the DNMT1, DNMT3A, and DNMT3B Genes

DNA methylation ensures chromatin stability, gene expression, and genetic imprinting. DNA methyltransferases (*DNMT*s), such as *DNMT1*, *DNMT3A*, and *DNMT3B*, are responsible for transcribing and controlling DNA methylation patterns in mammals [43]. Notably, *DNMT1* maintains the methylation pattern of parent cells by acting on hemimethylated DNA strands during DNA replication. *DNMT3A* and *DNMT3B*, de novo methyltransferases, add methyl radicals to CpG sites without prior methylation marks [44]. More recent research has suggested that *DNMT* deficiencies contribute to tumor development and progression, proving that epigenetic changes caused by *DNMT* abnormalities are linked to carcinogenesis [43,45]. DNA methylation is the epigenetic marker that has been the subject of most related research, and environmental variables may influence this marker. Among these elements, ultraviolet radiation has received less attention in this context. Although there is no doubt about the connection between UV radiation and DNA mutations, little is known about the connection between UV radiation and epigenetic mechanisms. The DNA methylation profile of epidermal cells obtained from the skin can be altered by UV exposure [46]. According to the applied wavelength, additional studies suggest that exposure to UV light might impair or restore the harmonious balance of epigenetics [46].

Our research explicitly revealed the significant beneficial effects of betanin treatment on downregulating the expression of *DNMT1*, *DNMT3A*, and *DNMT3B* in HaCaT cell cultures. After treatment with high concentrations of betanin, these target genes were downregulated, and a photoprotective effect was observed. Other authors have also reported the suppressive effects of betanin derived from *Beta vulgaris* L. on cancerous genes. For example, in cancer cell lines linked to esophagus, liver, breast, lung, and skin cancers, betanin has been shown to increase apoptosis and decrease cell proliferation, angiogenesis, and inflammation [15,18]. Additionally, several studies have demonstrated that by boosting the production of antioxidants and eliminating lipoperoxyl free radicals, betanin exhibits chemotherapeutic and antiproliferative properties [21,22]. Currently, available data indicate that the silencing of tumor suppressor genes causes photo carcinogenesis in the epidermis exposed to UV radiation and is linked to a network of epigenetic modifications, including changes in DNA methylation, the activity of DNA methyltransferases, and histone acetylation. In animal models, several bioactive dietary ingredients can decrease the risk of skin cancer induced by UV radiation [47]. Natural bioactive metabolites, especially those isolated from herbal plants, have shown anticancer effects linked to the modulation of DNMTs and their subsequent mechanisms of action [48].

De novo methyltransferases oversee the generation of DNA methylation patterns during embryogenesis and create genomic imprints during the formation of germ cells. Although *DNMT3A* and *DNMT3B* are abundantly expressed in early mammalian embryos, their expression levels decline as cells differentiate. These two proteins play different roles throughout embryonic development and exhibit location-based and momentary disparities [44]. Numerous studies have suggested a connection between overexpressed DNMTs and carcinogenesis. Zhao et al. demonstrated that the knockdown of DNMTs inhibits the cell cycle in esophageal squamous cell carcinoma [49]. Two studies have shown that *DNMT3B* overexpression and CIMP-high expression are closely associated with colon tumors [50,51]. Considering the above findings, we, therefore, present betanin as a promising photoprotective compound against the detrimental influence of UV irradiation exposure by effectively downregulating the expression of the *DNMT1*, *DNMT3A*, and *DNMT3B* genes.

### 4.2. Molecular Effects of Betanin on the HDAC5 and HDAC6 Genes

Histone deacetylases are evolutionarily invariant enzymes that eliminate acetyl groups from histones and other protein regulatory elements, resulting in physiological impacts on chromatin structure and transcription profiles [52]. The significance of *HDAC* enzymes in numerous biological processes, such as cell cycle regulation, proliferation, survival, differentiation, metabolism, protein trafficking, DNA repair, and angiogenesis, makes them an intriguing class of targets for pharmacotherapy. Natural phytoconstituents found in plants are being used to identify a class of inhibitors that have been seen in recent decades to regulate *HDAC* activity. Through the hyperacetylation of histones, these inhibitors can prevent the inactivation of genes. They may impact the regulation of gene expression, cell proliferation, and cell differentiation, as well as the activation of apoptosis in malignancies [53]. Therefore, understanding the complexities of the *HDAC* function in cells is critical for developing pharmaceutical techniques to inhibit or modulate the initiation and maintenance of pathogenesis. The class IIb deacetylase with the most excellent characterization is *HDAC6*, which controls various crucial biological processes by forming complexes with other proteins. Both cytoplasmic and nuclear processes depend on *HDAC6*. *HDAC6* has a distinct substrate specificity for nonhistone proteins compared to other deacetylases [54]. With such a broad spectrum of actions, *HDAC6* may be a valuable therapeutic target for many different disorders. The discovery that *HDAC6* may be overexpressed or dysregulated in various malignancies, neurological illnesses, and inflammatory disorders has attracted therapeutic interest [54]. The *HDAC5* gene belongs to the class II HDAC alpha family, and the protein has diverse substrates. This gene has two transcript variants that encode two distinct isoforms. *HDAC5* exhibits histone deacetylase activity when bound to a promoter and represses transcription. Other authors demonstrated that knocking down *HDAC5* or *HDAC6* inhibited melanoma cell proliferation and promoted apoptosis [55].

Beetroot extracts have been reported in recent studies to have cytotoxic activity in cancer cells and to play a preventative function against cancer. These extracts included a wide range of natural chemicals, including betanin and its stereoisomer isobetanin, which are highly bioavailable antioxidants that are part of the betalain group. However, it is still necessary to identify the exact chemicals causing this tumor-inhibitory impact. Crucially, normal cells do not exhibit any overt effects from the betanin extract [56], which underscores their significance in contrast to synthetic chemotherapeutic treatments empirically in use, that exhibit adverse side effects. Our findings demonstrated that betanin is an effective inhibitor of *HDAC5* and *HDAC6* and has significant beneficial effects. The photoprotective effects of betanin treatment through the observed decrease in the expression of *HDAC*s in our study indicate the importance of betanin as a promising inhibitor for postexposure photoprotection upon UV irradiation. Also documented are betanin’s additional chemopreventive effects, which include a persistent delay in tumor onset, an increase in tumor latency, a reduction in tumor multiplicity, and a further decrease in splenomegaly [19,20]. 

Furthermore, treatment with 40 mM betanin of the human chronic myeloid leukemia cell line-K562 has been demonstrated to induce an intrinsic apoptosis pathway by activating caspase-3, an executioner caspase in apoptotic cascades, and to reduce cell proliferation by 50% [57,58]. Beetroot is recommended for the prevention and management of cancer’s metastatic progress in traditional Persian medicine as well as other medical systems such as Arab, traditional Chinese, and Ayurvedic medicine [59]. Additionally, patients with breast, prostate, and colorectal cancer in Trinidad and Tobago use beetroot as a functional food and herbal remedy [60]. In certain cultures and nations, beetroot is consumed by most patients with gastrointestinal cancer in order to reduce the side effects of chemotherapy and as an alternative dietetic measure. From these significant findings, we, therefore, conclusively present betanin as a promising therapeutic inhibitor of *HDAC5* and *HDAC6.*

### 4.3. Betanin Ameliorates UV-Related DNA Fragmentation

This study assessed DNA fragmentation and found that betanin treatment resulted in a dose-dependent improvement in UV-irradiated cells. Our results suggest that betanin prevents DNA damage in oxidative stress conditions by deactivating the enzymes responsible for the hypermethylation and deacetylation of chromatin.

UV radiation is widely recognized for stimulating oxidative stress by generating reactive oxygen species (ROS) and initiating cellular transcriptional responses. This process encompasses the activation of factors such as NF-κB, AP-1, and MAPK, as well as the induction of STAT3 tyrosine phosphorylation. Consequently, this cascade augments the synthesis of inflammatory mediators and ultimately contributes to the progression of cellular death [61]. In situations where the generation of ROS surpasses cellular antioxidant defense, oxidative stress can inflict harm on DNA, proteins, and lipids [62]. Studies have indicated that betanin hinders the peroxynitrite-induced processes of tyrosine nitration and DNA strand breakage [63]. In a particular study, the researcher subjected HT-29 cells to hydrogen peroxide to initiate DNA damage facilitated by ROS. The extent of DNA damage was assessed using the comet assay. Prior treatment of HT-29 cells with betanin markedly mitigated the DNA damage induced by H_2_O_2_ [22]. The study’s findings align with this present study’s results since beetroots are characterized using distinctive antioxidants, with betalains, particularly red betacyanin–betanin, being the predominant compound [64].

## 5. Conclusions

This study is the first to explore the impact of betanin on DNMT and HDAC genes in HaCat cells. The research reveals that betanin can alter the gene expressions of DNMT1, DNMT3a, DNMT3b, HDAC5, and HDAC6 with and without UVB radiation. Based on the data, we hypothesize that betanin may act as an antioxidant and photo-protective agent against UVB radiation in HaCaT cells, compared to previous research findings.

## Figures and Tables

**Figure 1 nutrients-16-00860-f001:**
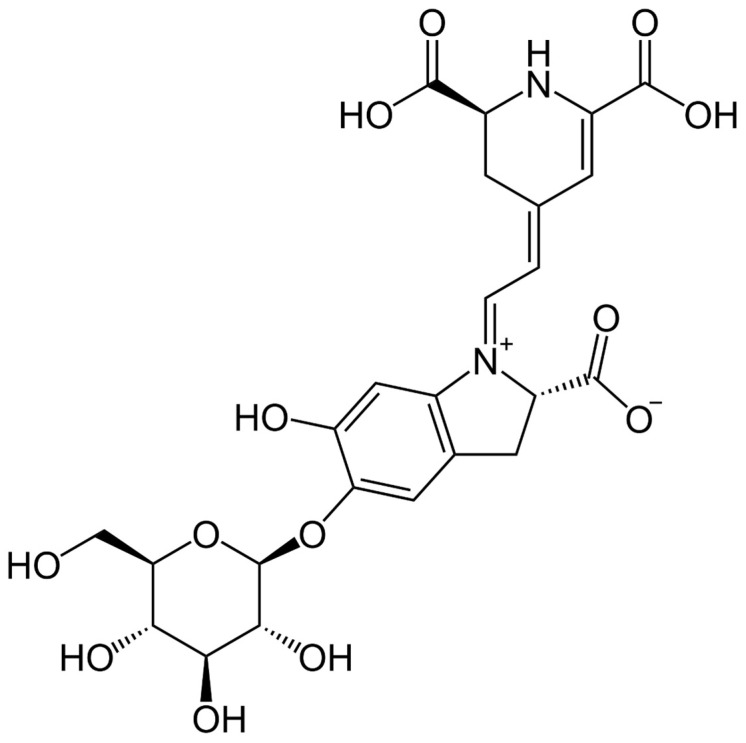
Chemical structure of betanin.

**Figure 2 nutrients-16-00860-f002:**
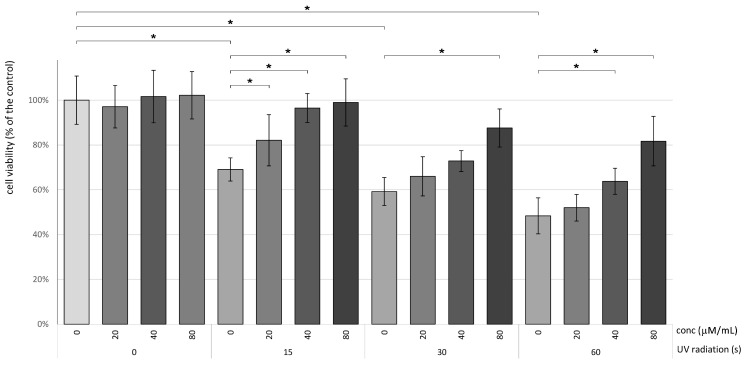
The impact of betanin on the viability of UVB-irradiated human keratinocytes (HaCaT) was assessed. Following a 24 h treatment period, betanin resulted in a notable augmentation in keratinocyte viability, as confirmed by the MTT assay. Data are presented as mean ± SD and were statistically analyzed using one-way ANOVA with Tukey’s post hoc test. The experiments were conducted in triplicate, with significance denoted by * at *p* < 0.05.

**Figure 3 nutrients-16-00860-f003:**
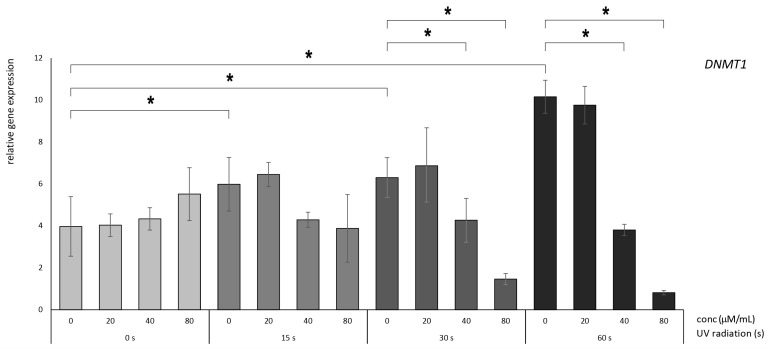
Relative gene expression level of DNMT1 in HaCaT cells, *p* < 0.05, after UV radiation for 15, 30, or 60 s, followed by betanin treatment at concentrations of 20, 40, and 80 µM (* = *p* < 0.05).

**Figure 4 nutrients-16-00860-f004:**
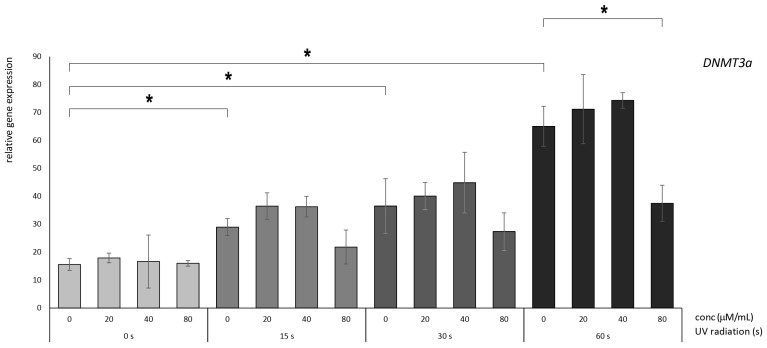
Relative gene expression level of DNMT3A in HaCaT cells, *p* < 0.05, after UV radiation for 15, 30, or 60 s, followed by betanin treatment at concentrations of 20, 40, and 80 µM (* = *p* < 0.05).

**Figure 5 nutrients-16-00860-f005:**
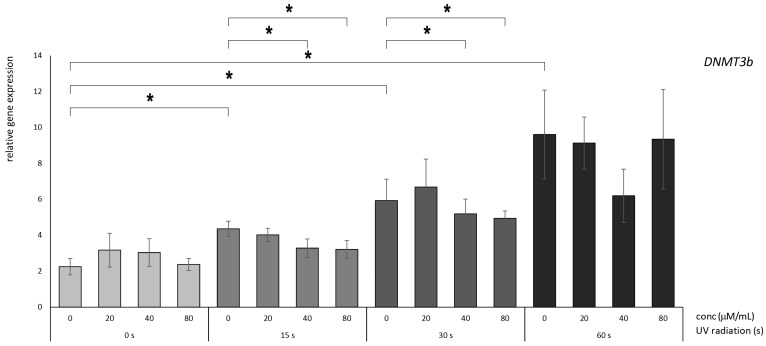
Relative gene expression level of DNMT3B in HaCaT cells, *p* < 0.05, after UV radiation for 15, 30, and 60 s, followed by betanin treatment at concentrations of 20, 40, and 80 µM (* = *p* < 0.05).

**Figure 6 nutrients-16-00860-f006:**
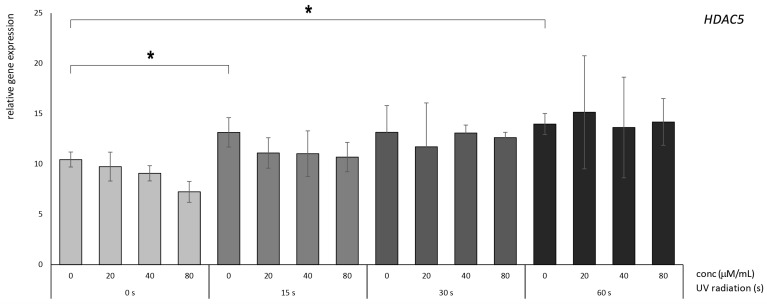
Relative gene expression level of HDAC5 in HaCaT cells, *p* < 0.05, after UV radiation for 15, 30, and 60 s, followed by betanin treatment at concentrations of 20, 40, and 80 µM (* = *p* < 0.05).

**Figure 7 nutrients-16-00860-f007:**
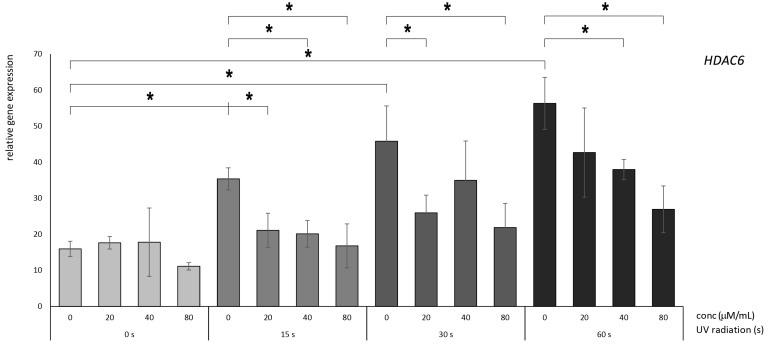
Relative gene expression level of HDAC6 in HaCaT cells, *p* < 0.05, after UV radiation for 15, 30, and 60 s, followed by betanin treatment at concentrations of 20, 40, and 80 µM (* = *p* < 0.05).

**Figure 8 nutrients-16-00860-f008:**
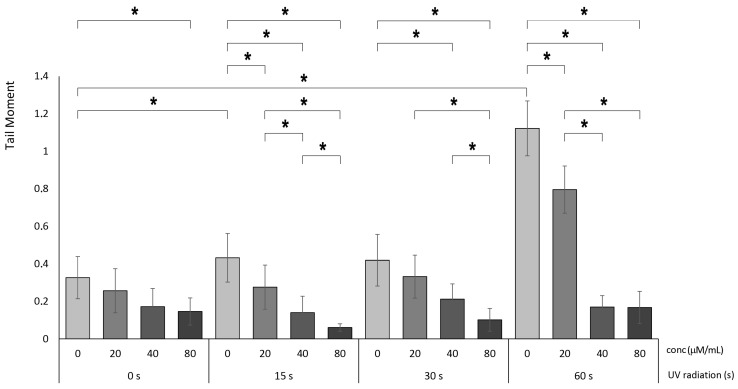
DNA fragmentation (tail moments) in HaCaT cells, *p* < 0.05, after UV radiation for 15, 30, and 60 s, followed by betanin treatment at concentrations of 20, 40, and 80 µM (* = *p* < 0.05).

**Table 1 nutrients-16-00860-t001:** Betanin treatment protocol.

	Name of the Groups	UV Radiation (s)	Food Colorant (µM)
1	neg. control	0	0
2	treated #1	0	20
3	treated #2	0	40
4	treated #3	0	80
5	treated #4	15	20
6	treated #5	15	40
7	treated #6	15	80
8	treated #7	30	20
9	treated #8	30	40
10	treated #9	30	80
11	treated #10	60	20
12	treated #11	60	40
13	treated #12	60	80
14	pos. Control #1	15	0
15	pos. Control #2	30	0
16	pos. Control #3	60	0

**Table 2 nutrients-16-00860-t002:** Sequences of primers used for relative gene expression level measurement via qRT–PCR.

Gene	Forward Primer	Reverse Primer
*DNMT1*	5′-AGGTGGAGAGTTATGACGAGGC-3′	5′-GGTAGAATGCCTGATGGTCTGC-3′
*DNMT3A*	5′-GCA GCG TCA CAC AGA AG-3′	5′-GGC GGT AGA ACT CAA AGA AG-3′
*DNMT3B*	5′-GAA CGA CGT GAG GAA CAT C-3′	5′-GGC CTG TAC CCT CAT ACA-3′
*HDAC5*	5′-CAG CAC CAT CGG TTC ATA G-3′	5′-CAG GGA GAG AGT GGG TAA G-3′
*HDAC6*	5′-GCC CAG GCT TCA GTT TC-3′	5′-CCT CGC TCT CCT CTA CAT T-3′
*HPRT1*	5′-TGC TTC TCC TCA GCT TCA-3′	5′-CTC AGG AGG AGG AAG CC-3′

## Data Availability

Data will be made available upon request due to that this manuscript is part of a multiple-study.

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
