# Peer review of "Betanin Attenuates Epigenetic Mechanisms and UV-Induced DNA Fragmentation in HaCaT Cells: Implications for Skin Cancer Chemoprevention"

_nutrients, 2024, doi:10.3390/nu16060860_

Round 1
Reviewer 1 Report
Comments and Suggestions for Authors
Dear Authors,
The work was well developed, but it can be improved. Some points need to be revised.
- There is an excess of citations and references in the text present in the introduction. In total, more than 50 references were inserted in the text and this made the manuscript a little tiring. In my opinion, part of the introduction could be moved to the discussion. Also, the introduction should be shortened to be more concise and clear.
- the results, despite being modest, have scientific value. In the highest concentrations used, the natural substance presented activity.
- the betanin structure should be inserted into the text to make it more interesting for readers;
- the text of the results were well presented, however they lack discussion based on the natural substance and information about betaine related to the effects of enzyme inhibition that can be found in other works and found in other species of plants as well. It would enrich the work.
- Pls: check all the references. They are not according to the journal rules!
Author Response
Response to Reviewer’s Comments
Dear Editor,
My co-authors and I deeply value the encouraging, critical, and constructive feedback on this manuscript provided by the reviewer. The thoroughness and utility of the comments have contributed significantly to enhancing the quality of our work. We firmly believe that incorporating these suggestions has substantially elevated the revised manuscript's scientific merit. We have diligently integrated them into the revision process and are now submitting the corrected manuscript with the recommended changes implemented. Our revisions align closely with the reviewer's comments, and we provide detailed responses to each of them below:
Reviewer #1:
- There is an excess of citations and references in the text present in the introduction. In total, more than 50 references were inserted in the text and this made the manuscript a little tiring. In my opinion, part of the introduction could be moved to the discussion. Also, the introduction should be shortened to be more concise and clear..
Response- Thank you so much for your suggestion. Yes, we have Indeed, we have condensed the introduction section and minimized the number of references to enhance its readability and engagement.
- The results, despite being modest, have scientific value. In the highest concentrations used, the natural substance presented activity.
Response- Thank you so much for your minute observation and valuable comments.
- the betanin structure should be inserted into the text to make it more attractive for readers.
Response- Thank you very much for your valuable suggestion. We have taken reviewer’s comment The chemical structure of betanin has been incorporated into our manuscript.
- the text of the results were well presented, however they lack discussion based on the natural substance and information about betaine related to the effects of enzyme inhibition that can be found in other works and found in other species of plants as well. It would enrich the work.
Response- Thank you so much for your comment. We have expanded the discussion section and added 14 additional references to enrich the manuscript further.
- Pls: check all the references. They are not according to the journal rules!
Response- Thank you very much for your valuable suggestion, For this manuscript, we utilized the Mendeley reference manager, ensuring that all references were cited according to the citation style provided by the Multidisciplinary Digital Publishing Institute.

Reviewer 2 Report
Comments and Suggestions for Authors
The authors of this submission reported the effect of betanin on the epigenetic mechanisms and UV-induced DNA fragmentation in HaCaT cells. In my opinion, although the authors have made an effort in presented work, the manuscript must be improved in terms of organization scientific experiments and scientific rigor. In this form should not be accepted for publication before it’s major revision.
1) There is no description of the used materials.
2) There is no information about the percent of confluency of HaCaT cells. These type cells are too sensitive to cell density. The seeding cells at a density 0.5 x 10^6 cells per well is too crowded. The information about the confluency and microscopic pictures of the cells before and after experiment need to be applied.
3) It is not clear when exactly UVB radiation start, how HaCaT cells are cultivated before experiment and when cells are seeded in 6-well plate.
4) More experiments about cell cytotoxic analysis and ROS generation need to be performed.
Comments on the Quality of English Language
Minor editing of English language required.
Author Response
Response to Reviewer’s Comments
Dear Editor,
The constructive feedback offered by the reviewer on this manuscript is greatly appreciated by myself and my co-authors. We recognize the thoroughness and usefulness of the comments, which have significantly improved the quality of our work. We are confident that incorporating these suggestions has notably enhanced the scientific value of the revised manuscript. These recommendations have been carefully integrated into our revision process, and we are now submitting the corrected manuscript with the suggested changes applied. Our revisions closely adhere to the reviewer's comments, and detailed responses to each are provided below:
Reviewer #2:
- There is no description of the used materials.
Response- Thank you so much for your comment. We have meticulously integrated all material descriptions into the manuscript.
- There is no information about the percent of confluency of HaCaT cells. These type cells are too sensitive to cell density. The seeding cells at a density 0.5 x 10^6 cells per well is too crowded. The information about the confluency and microscopic pictures of the cells before and after experiment need to be applied.
Response- Thank you so much for your minute observation and valuable comments. We identified an error in the information provided regarding the passage and seeding of the HaCaT cell line. We have rectified all errors and completely rewritten this section for clarity. In considering the inclusion of microscopic images, after extensive discussion and debate among the authors, it was decided that including such images would require adding 30 photos, potentially making the manuscript overly burdensome.Please refer to the attached files for microscopic images of both treated and untreated HaCaT cells.
- It is not clear when exactly UVB radiation start, how HaCaT cells are cultivated before experiment and when cells are seeded in 6-well plate.
Response- Thank you for your comment. We have completely revised the materials and methods section, offering a clearer description of the experiments. Additionally, we have included an extra table to enhance reader engagement.
- More experiments about cell cytotoxic analysis and ROS generation need to be performed.
Response- Thank you for your feedback. We conducted the XTT assay, a valuable method for measuring cell viability, which has been statistically evaluated and included in the manuscript. In terms of ROS generation, we referenced several articles that have utilized this assay and incorporated their findings into our discussions.
